# Electrostatics in semiconducting devices II : Solving the Helmholtz equation

**Antonio Lacerda-Santos [1,2]⋆ and Xavier Waintal [1]†**

**1** Univ. Grenoble Alpes, CEA, IRIG-PHELIQS GT, F-38000 Grenoble, France **2** CEA Saclay, IRAMIS-SPEC GMT , CEDEX 91191 GIF-SUR-YVETTE, France

⋆ antoniosnl@hotmail.com , † xavier.waintal@cea.fr

## Abstract

The convergence of iterative schemes to achieve self-consistency in mean field problems such as the Schrödinger-Poisson equation is notoriously capricious. It is particularly difficult in regimes where the non-linearities are strong such as when an electron gas in partially depleted or in presence of a large magnetic field. Here, we address this problem by mapping the self-consistent quantum-electrostatic problem onto a Non-Linear Helmoltz (NLH) equation at the cost of a small error. The NLH equation is a generalization of the Thomas-Fermi approximation. We show that one can build iterative schemes that are *provably* convergent by constructing a convex functional whose minimum is the seeked solution of the NLH problem. In a second step, the approximation is lifted and the exact solution of the initial problem found by iteratively updating the NLH problem until convergence. We show empirically that convergence is achieved in a handfull, typically one or two, iterations. Our set of algorithms provide a robust, precise and fast scheme for studying the effect of electrostatics in quantum nanoelectronic devices.

# 1   Introduction

This article is the second of a series of three articles centered around the numerical solution
of the self-consistent quantum-electrostatic problem that occurs in quantum nanoelectronic
devices. The first article of the series discusses a minimum approximation level that captures
the gross features of the problem and allows for the identification of the small parameter of
the problem [1] . The present article focusses on a set of robust algorithms to achieve self-
consistency. The last article describes the PESCADO open source library that implements the
algorithms described below [2].

## 1.1   Continuous problem definition

The self consistent quantum electrostatic problem (SCQE), also known as the Schrödinger-
Poisson problem or the self consistent Hartree problem, is the minimum mean field approach
that describes the quantum mechanics of electrons subject to their own electric field. It con-
sists of a cascade of three equations that describe respectively quantum mechanics, statistical
physics and electrostatics. First comes the Schrödinger equation :

$$[H_0 - eU]\Psi_{\alpha E} = E\Psi_{\alpha E} \tag{1}$$

where $H_0$ is the system Hamiltonian, $U(\vec{r})$ the electrostatic potential seen by the carriers (that
we suppose to be electrons with negative charges without loss of generality), $e > 0$ the ele-
mentary charge, $\Psi_{\alpha E}$ the electronic wave function at energy $E$ and sub-band $\alpha$. In the simplest
situation $H_0 = p^2/(2m^*)$ is just the effective mass Hamiltonian but it can also account for or-
bital degrees of freedom and/or spins, superconductivity etc. We suppose that the system is in
the thermodynamic limit so that the energy $E$ varies continuously. The index $\alpha$, however, is dis-
crete and labels the different sub-bands (including those originating from the non-trivial band
structure and those originating from spatial confinement). A typical setup that corresponds to
the above is a finite central conductor connected to semi-infinite quasi-one dimensional elec-
trodes as studied in coherent quantum transport [3]. This is the setup adopted in the Kwant
package [4] which we have used in our quantum calculations. Solving the quantum problems
provides the wavefunction $\Psi_{\alpha E}(\vec{r})$ or more generally the Local Density of State (LDOS) that
counts the density of electrons per unit volume and per unit energy,

$$\rho(\vec{r}, E) = \frac{1}{2\pi} \sum_\alpha |\Psi_{\alpha E}(\vec{r})|^2 \tag{2}$$

Filling the states up to the Fermi energy (statistical physics) provides the electron density $n(\vec{r})$,

$$n(\vec{r}) = \int dE \rho(\vec{r}, E) f(E) \tag{3}$$

where $f(E) = 1/[e^{\beta(E-\mu)} + 1]$ is the Fermi function with $\mu$ the electro-chemical potential and $\beta = 1/k_B T$ the inverse temperature. This equation can also be straightforwardly generalized to stationary non-equilibrium situations with bias voltages across different electrodes [3] (see also the first article of this series [1]). At zero temperature, to which we focus on this article for definiteness, Eq.(3) simplifies into

$$n(\vec{r}) = \int_{-\infty}^{\mu} dE \rho(\vec{r}, E) \tag{4}$$

where the left hand side is referred to as the integrated local density of states (ILDOS). Both the LDOS and the ILDOS are, implicitly, functionals of the electric potential $U$. The last equation is simply the Poisson equation

$$\nabla \cdot (\epsilon(\vec{r}) \nabla U(\vec{r})) = en(\vec{r}) - en_d(\vec{r}). \tag{5}$$

where $\varepsilon(\vec{r})$ is the dielectric constant and the doping charge density $n_d(\vec{r})$ accounts for electric charges that are not treated at the quantum mechanical level (for instance dopants or charges trapped at a surface). The Poisson equation is complemented by boundary conditions such as Dirichlet $U(\vec{r}) = V_g$ at metallic gates or Neumann on other boundaries.

The goal of this article is to design a robust, precise and fast technique to solve the SCQE problem in presence of strong non-linearities.

## 1.2 On the importance of the SCQE problem

Nature rarely accumulates significant amounts of electric charge across large distances. The reason is simply that the electric interaction is very strong so that it is usually energetically favorable to keep electro-neutrality down to the smallest - atomic or molecular - scale. There are a few exceptions, however, such as solutions or electrolytes, the charges across a capacitor or a super capacitor and the subject of this article, low dimensional nanoelectronic devices. These systems include semi-conducting quantum wells, field effect transistors, graphene or other van der Waas devices, nanowires. The importance of the SCQE problems stems from the fact that the first step in understanding these devices is to determine where the charges lie and the potential seen by the electrons [5]. Interestingly, this step is often bypassed in many quantum transport studies where one simply assumes a shape of the electric potential (for instance flat with hard walls on the boundaries of the sample) and the corresponding electronic density [6]. Such an approach can be sufficient to predict *some* features but fail at others. An obvious example is that SCQE calculations are needed to calculate the conductance versus gate voltages characteristics i.e. versus the actual knobs controled experimentally. Another example is the electrostatic reconstruction taking place in the edge states of the quantum Hall effect which is absent from the simplest models that ignore the SCQE effects [7,8].

The basic approach to solve SCQE problems is to use some sort of iterative algorithm that starts from an initial guess of the electrostatic potential, solves the Schrödinger equation, fills the orbitals to get the density, solve the Poisson equation and then repeat the procedure until convergence [9–12]. SCQE problems fall into two categories: confined systems (e.g. molecules) described by a discrete gapped set of orbitals and extended systems (the subject of this work) with a continuous spectrum. In the former, iterative algorithms usually converge rather well. The same is not true for metallic extended systems, where iterative algorithms tend to diverge. In metallic extended systems an additional difficulty comes from the fact that not only do orbitals depend on the electrostatic potential, but their filling does too, hence one can observe strong variations from one iteration to the next - rendering convergence difficult to achieve.

To improve on the convergence of iterative algorithms one of the simplest method is the under-relaxation algorithm [13,14]. It consists of adding a damping parameter on the previous iterative solution. Such a step may help but even when simple iterations with or without under-relaxation converge, they often do it slowly [15]. More stable and faster approaches tend to derive the next iteration input by mixing the solutions of several previous iterations [16]. For instance, the Anderson mixing uses a linear combination of several previous iteration as the next iteration input [17–20]. More advanced mixing methods include the Direct Inversion in the Iterative Subspace, a.k.a. Pulay mixing [21] and the Broyden mixing [22].

Besides direct iterative algorithms, fast convergence can be obtained (in some regimes) using root-finding methods. Prime examples are Newton-Raphson based algorithms [23–26]. The major improvement that they bring is that the iterations are not based solely on the density $n(\vec{r})$ but also on its (approximate) derivative with respect to energy. In other words, the self-consistent algorithm uses information about the existence of a Fermi level. More modern methods implement a Predictor-Corrector approach [27–31] that generalize simple root finding schemes. The present article can be considered as further generalisation of these schemes where the self-consistency is achieved through iteratively updating the *full* ILDOS (i.e. the density versus *both* space and electro-chemical potential).

Indeed, a notable limitation oberved in Predictor-Corrector approaches is lack of convergence in presence of rapid variations of the LDOS with space (e.g. at a depletion edge) and/or energy (at e.g. the onset of a band) where the ILDOS has cusps. This is due to these methods being often based on a linearisation of the ILDOS with respect to energy, which is problematic when the non-linearities are strong. The method proposed in this paper explicitly handles strong non-linearities including the presence of cusps. It is the natural extension of [32] that places the long range aspect of the problem on the same footing as the non-linear aspect.

This paper is organized as follows. We first formulate in Section 2 a discrete version of the SCQE problem. Then, in Section 3 we map (approximately) the SCQE discrete problem onto a non-linear Helmholtz (NLH) equation. We show that the NLH equation can be solved with provable convergence using a convexity argument. In Section 4 we discuss two practical algorithms for solving the NLH equation, which we respectively call Piecewise Newton-Raphson and Piecewise Linear Helmholtz. These two algorithms are illustrated in Section 5 for a particular example, a nanowire with an hexagonal section. Finally, we show in Section 6 how to relax the approximation of the NLH equation and obtain the converged solution of the original SCQE problem.

## 2    Discrete problem definition

We start with formulating a discrete version of the SCQE problem for both the quantum problem and the electrostatic problem, suitable for numerical computations.

The Poisson equation can be discretized in many ways using e.g. finite difference or finite elements. Here we use finite volumes which has the advantage that for any discretization, however coarse, we have a valid discrete electrostatic problem defined in terms of a capacitance matrix $C_{ij}$. We discretize the simulation volume into a set of cells $\mathcal{C}_i$ centered on point $\vec{r}_i$. Each cell has a few neighbors $j$ at distance $d_{ij} = |\vec{r}_i - \vec{r}_j|$ and $S_{ij}$ is the surface that separates them. We use Voronoi cells so the surface is planar. Gauss theorem for a given cell takes the form,

$$\sum_j \Phi_{ij} = -eQ_i \tag{6}$$

where $Q_i$ is the total number of charge inside the cell ($Q_i = -1$ for one electron $+1$ for one

hole),

$$Q_i(\mu) = -\int_{\mathcal{C}_i} d\vec{r} \, [n(\vec{r}) - n_d(\vec{r})], \tag{7}$$

and $\Phi_{ij}$ the flux of the electric field

$$\Phi_{ij} = \int_{S_{ij}} \varepsilon(\vec{r}) \vec{\nabla} U(\vec{r}) \cdot \vec{n} \, \mathrm{d}S \tag{8}$$

where $\vec{n}$ is the unit vector perpendicular to the surface $S_{ij}$ (parallel to $\vec{r}_i - \vec{r}_j$ for Voronoi cells). We approximate $\vec{\nabla} U(\vec{r}) \cdot \vec{n}$ to first order with $(U_j - U_i)/d_{ij}$ where $U_i$ the electric potential at the center of cell $i$. We arrive at

$$\sum_j C_{ij} U_j = Q_i(\mu) \tag{9}$$

with the capacitance matrix given by

$$C_{i \neq j} = -\frac{\varepsilon_{ij} S_{ij}}{e d_{ij}} \leq 0 \tag{10}$$

for neigbouring cells,

$$C_{ii} = -\sum_{j(i)} C_{ij} \geq 0 \tag{11}$$

for the diagonal part ($j(i)$ stands for the neighbors of cell $i$) and $C_{ij} = 0$ otherwise (beware of the slight abuse of notations since we have incorporated the electric charge $e$ inside the definition of $C_{ij}$). The dielectric constant $\varepsilon_{ij}$ is averaged over neighboring sites according to

$$\varepsilon_{ij} = \frac{2\varepsilon_i \varepsilon_j}{(\varepsilon_i + \varepsilon_j)} \tag{12}$$

where $\varepsilon_i$ is the dielectric constant inside cell $i$. Finally, defining the discrete LDOS as

$$\rho_i(E) = \int_{\mathcal{C}_i} d\vec{r} \, \rho(\vec{r}, E), \tag{13}$$

we have for electrons

$$\frac{\partial Q_i}{\partial \mu} = -\rho_i(\mu) \leq 0. \tag{14}$$

As for the quantum problem, there are also various ways to obtain a discretized model, e.g. tight-binding model, finite difference from a k.p Hamiltonian etc. Here we suppose that this discretized model is obtained on the same cartesian grid as the electrostatic model, possibly with additional local degrees of freedom on each site (spin, orbitals, different atoms per unit cell etc.) in the usual framework of quantum transport [4]. $\Psi_{\alpha E}$ becomes a vector with $\Psi_{\alpha E}(i)$ the subvector on site $i$ (whose components span the local degrees of freedom). The discrete Schrödinger equation reads

$$\sum_j [(H_0)_{ij} - eU_i \delta_{ij} \mathbb{1}] \Psi_{\alpha E}(j) = E \Psi_{\alpha E}(i) \tag{15}$$

where $\mathbb{1}$ is the identity matrix of the local degrees of freedom. With these notations, the discrete LDOS reads,

$$\rho_i(E) = \frac{1}{2\pi} \sum_\alpha \Psi_{\alpha E}(i)^\dagger \Psi_{\alpha E}(i). \tag{16}$$

Together, the above set of equations forms the discrete SCQE problem.

# 3  The Non-Linear Helmholtz (NLH) problem

The difficulty of the SCQE problem stems from the combination of two factors: the electrostatic problem is non-local (long range) and quantum mechanics gives rise to potentially strong non-linearities (the density, which enters the electrostatic problem, is the square of the wave-function). A typical iterative scheme addresses the non-local aspect well but treats the non-linearity badly, relying on good approximate solutions for convergence. In [32], we used a scheme that treated the non-linearity exactly but relied on an iterative scheme to address the non-local aspects. The scheme proved to be very good on some problems, but failed in others. In this article, we improve on [32] by introducing a scheme that solves an approximate non-local non-linear problem exactly (with provable fast convergence) and then relax the approximation iteratively. Our numerics suggest that the convergence is obtained in a handful of iterations, often just one or two are sufficient.

In this section we address three tasks in three subsections: first, we introduce the quantum adiabatic approximation (QAA) which is valid in the limit where the electrostatic potential varies slowly with respect to the characteristic scales of the quantum problem. QAA essentially supposes that the LDOS $\rho_i(E)$ is independent of the electrostatic potential up to a simple shift. Under QAA, the SCQE problem is mapped onto a generalized non-linear Helmholtz (NLH) equation. Second, we show that the NLH equation can be solved efficiently with provable unconditional convergence. We will discuss several algorithms to solve NLH equation in practice in the next section. Third, we show how the NLH solver can form the basis of a very robust SCQE solver.

## 3.1  The Quantum Adiabatic Approximation

QAA is a generalization of the Thomas-Fermi approximation, it has been discussed in detail in [32]. Suppose that for an electrostatic potential $U_i$ we have computed the LDOS $\rho_i(E)$. Now consider a different potential $U_i' = U_i + \delta U_i$. QAA approximates the corresponding LDOS as,

$$\rho_i'(E) = \rho_i(E + e\delta U_i) \tag{17}$$

i.e. it supposes that the electrostatic potential simply shifts the different energy bands. The approximation is exact in the limit where the difference of potential $\delta U_i$ varies infinitely smoothly with position. Noting $E_F$ the electro-chemical potential of the quantum problem, Equation (9) reduces to an equation for $U_i'$

$$\sum_j C_{ij} U_j' = Q_i(E_F + eU_i' - eU_i) \tag{18}$$

which is the Non-Linear Helmholtz equation mentioned earlier. Eq.(18) can be recast into a slightly more convenient form,

$$\sum_j C_{ij}\delta U_j = Q_i(E_F + e\delta U_i) + n_i \tag{19}$$

where the source term $n_i$ is given by $n_i = -\sum_j C_{ij} U_j$. Hereafter, we ignore the $n_i$ term that we absorb in the definition of $Q_i$.

In contrast to the SCQE problem for which, as far as we are aware, there is no proof of convergence of the various iterative schemes that are commonly used, we shall see that the NLH equation has very nice properties both theoretically and in practice. For small $\delta U_i$, we can linearize the equation and obtain a discretized version of the (Linear) Helmholtz equation (LH),

$$\sum_j C_{ij}\delta U_j = Q_i(E_F) - e\rho_i(E_F)\delta U_i. \tag{20}$$

The LH equation is a (sparse) linear equation that can be solved by standard numerical approaches (the density of state term is moved to the left hand side). All the algorithms described in this article eventually reduce to sequences of calls to the LH equation solver. Last, if $U_i = 0$ and the LDOS is the bulk DOS $\rho^{\text{bulk}}$ at the Fermi energy (site independent) then the LH equation reduces to (a generalization of) the well-known Thomas Fermi approximation [20, 27, 33, 34],

$$\sum_j C_{ij} U_j = Q_i(E_F) - e\rho^{\text{bulk}} U_i. \tag{21}$$

## 3.2 The NHL equation as the minimum of a convex functional

The advantage of using the NLH equation stems from the fact that it can be proved that it is free from the convergence problems that plagues the original SCQE problem. We provide the corresponding argument in this section.

The NHL equation (18) thus takes the generic form,

$$\sum_j C_{ij} U_j = Q_i(U_i) \tag{22}$$

with the following properties:

- $C_{ij}$ is symmetric.

- $C_{ij}$ is semi-definite positive. Indeed, $\sum_j C_{ij} = 0$ and $C_{i\neq j} \leq 0$ imply that $\forall U_i$,

$$\sum_{ij} U_i C_{ij} U_j = -\frac{1}{2} \sum_{i\neq j} (U_i - U_j)^2 C_{ij} \geq 0 \tag{23}$$

- $Q_i(E)$ is a decreasing function of the energy ($dQ_i/dE = -\rho_i(E)$) and the LDOS is positive).

- $\rho_i(E) = 0$ for $E < E_B$, the bottom of the lowest band.

The goal of this section is to construct a functional $F(\{U_i\})$ that (i) admits the solution of the NLH equation as its global minimum and (ii) has no local minimum or saddle points. The existence of such a functional guarantees the unconditional convergence of a scheme where one would solve the NHL by minimizing $F$.

We define $F(\{U_i\})$ as,

$$F(\{U_i\}) = \frac{1}{2} \sum_{ij} U_i C_{ij} U_j - \int_{-\infty}^{U_i} dE \, Q_i(E). \tag{24}$$

$F$ is the sum of two convex functions and is therefore convex itself. The gradient of $F$ is given by

$$\frac{\partial F}{\partial U_i} = \sum_j C_{ij} U_j - Q_i(U_i), \tag{25}$$

and its Hessian is,

$$\frac{\partial^2 F}{\partial U_i \partial U_j} = C_{ij} + \rho_i(U_i)\delta_{ij} \tag{26}$$

i.e. a semi-definite positive matrix (sum of two semi-definite matrices). This functional admits a global minimum which is the solution of the NLH equation. In the trivial case where all

bands are empty, NHL can have degenerate global minimums that differ by a global shift of the potential $U_i \to U_i + U$. However, when at least one band is partially occupied, one can explicitly check that there is a unique global minimum. The proof goes as follows: if $U_i^*$ is a minimum of $F$ then for any variation $\delta U_i^*$ around this minimum,

$$F(\{U_i^* + \delta U_i^*\}) = F(\{U_i^*\}) + \frac{1}{2} \sum_{ij} \delta U_i^* C_{ij} \delta U_j^*$$

$$-\int_{U_i^*}^{U_i^* + \delta U_i^*} dE \; [Q_i(E) - Q_i(U_i^*)] \tag{27}$$

which is the sum of two positive terms. $F(\{U_i^* + \delta U_i^*\}) = F(\{U_i^*\})$ implies that $\delta U_i^*$ does not depend on $i$ (first term) and the local density of states vanish on all sites at $E = U_i$, which contradict our hypothesis.

To summarize, we are in a very comfortable situation to solve the NLH equation: it is the global minimum of a convex functional of which we know both the gradient and the Hessian. Hence, any gradient descent approach must converge to the unique solution. This is a much more satisfactory situation than the SCQE problem we started with.

### 3.3 Overall scheme for solving the SCQE problem

Assuming that we have a robust NHL solver (the corresponding algorithms are discussed in the next section), we can build the solution of the SCQE problem iteratively as follows.

1. One starts with an initial LDOS $\rho_i(E)$. A common choice is to use the bulk DOS of the material on all sites $\rho_i(E) = \rho^{\text{bulk}}(E)$. Alternatively, one can solve the quantum problem with $U_i = 0$ on all sites.

2. Given this LDOS, one solves the NLH problem and obtain $U_i$.

3. Given this potential $U_i$, one solves the quantum problem and obtain a new LDOS.

4. One repeats steps (2) and (3) until convergence (no mixing scheme has been necessary in our experience so far).

The main difference between this strategy and standard iterative schemes is that the input of the electrostatic problem is not the density but the LDOS, i.e. a quantity that *contains information* about the energy dependance of the quantum problem. Indeed, the NLH equation already captures the main sources of non-linearities of the problem. In particular, it *knows* about the LDOS at the Fermi level, about the potential presence of gaps in the spectrum etc. If one chooses the bulk DOS as the initialization, then the first solution of the NLH equation gives a generalization of the Thomas-Fermi potential of the problem.

## 4 Practical algorithms for solving the NLH equation

We now turn to the practical schemes used to solve the NLH equation. Despite the formal convexity proof, there remains one small but crucial difficulty that we must treat with care: the LDOS is usually a smooth function of energy but it has cusps or discontinuities at the band edges. To illustrate this problem, let's consider a simple quadratic Hamiltonian $H \propto p^2$. The corresponding DOS has the form $\rho(E) \propto E^{d/2-1}$ and has a discontinuous derivative ($d = 3$), is discontinuous ($d = 2$) and even diverges ($d = 1$) at the band edge $E = 0$. The existence

of these singular points make traditional gradient descent like methods unstable unless one treats these points explicitly. This is particularly true in low dimension.

We handle this difficulty by explicitly tracking the problematic points in energy on each site. Physically, it means that we are tracking the regions of space that are depleted (due to e.g. a gate) and those that are not. Below, we explain the algorithm to deal with this aspect. Once this is taken care of, we have found most approaches to converge quickly to the solution. We present two of them: the piecewise Newton-Raphson algorithm (a variation of the eponym algorithm) and the piecewise linear Helmholtz algorithm. The first is an heuristic with good practical convergence properties. The second is provably convergent.

## 4.1 Breaking the ILDOS into piecewise-smooth regions

We start by discussing a common feature of the two NLH solvers used to track the cusps/singularities at the band edges. The input of a NLH solver is the ILDOS $Q_i(E)$. On each site $i$, we break the energy regions into different intervals $[E_i^\alpha, E_i^{\alpha+1}]$ where the energies $E_i^\alpha$ mark the position of the (possibly) singular point in energy of the ILDOS ($E_i^0 = -\infty$ by convention). A different function $Q_i^\alpha(E)$ is used on each subinterval. These different subintervals generalize the Dirichlet and Neumann sites that were introduced in the first article of this series [1].

For example, in the case of the quadratic band, one would use $E^0 = -\infty$, $E^1 = 0$ and $E^2 = +\infty$. For $E \in [-\infty, 0]$, one has $Q_i^0(E) = 0$ while inside the band $E \in [0, +\infty]$, one uses $Q_i^1(E) \propto E^{d/2}$. As explained below, our NHL solvers explicitly track in which interval $\alpha(i)$ the solution lies for each site $i$.

## 4.2 Piecewise Newton-Raphson algorithm

We first describe the Piecewise Newton-Raphson algorithm, a simple adaptation of the Newton-Raphson algorithm that explicitly treat the cusps/discontinuity points that would be problematic for the standard Newton-Raphson algorithm. The algorithm works as follows:

1. We initialize the potential on all sites $U_i$. A typical choice is $U_i = 0$ everywhere. On each site, we identify the energy interval $\alpha(i)$ such that $U_i \in [E_i^\alpha(i), E_i^{\alpha(i)+1}]$.

2. We linearize the NLH equation at $U_i$ and form Eq.(20). This is a linear equation that can be solved e.g. with a sparse LU solver such as MUMPS [35, 36]. The solution to the NL problem is called $U_i'$.

3. for all sites, if $U_i' \in [E_i^\alpha(i), E_i^{\alpha(i)+1}]$ then we set $U_i \to U_i'$. However, if $U_i' \leq E_i^\alpha(i)$ then the corresponding point has switched to the previous branch (e.g. the corresponding site is depleted). We set $\alpha(i) \to \alpha(i) - 1$ and initialize $U_i$ to the middle of the previous interval ($U_i \to E_i^\alpha$ if $\alpha = 0$). Likewise, if $U_i' > E_i^{\alpha(i)+1}$ then we switch to the next branch. We set $\alpha(i) \to \alpha(i) + 1$ and $U_i$ to the middle of the next interval.

4. We repeat steps (2) and (3) until convergence.

Keeping track of the index $\alpha(i)$ of the solution on each site is the key to prevent the band edges from destabilizing the algorithm; they are the main source of non-linearities of the problem. For instance, as soon as a site is depleted, it is updated to a value of $\alpha$ where the LDOS vanishes and therefore it stops contributing to the screening of other charges.

## 4.3 Piecewise Linear Helmholtz algorithm

In practice the piecewise Newton-Raphson algorithm is very stable in almost all the situations that we have encountered. When the ILDOS is particularly non-linear, it can nevertheless fail to

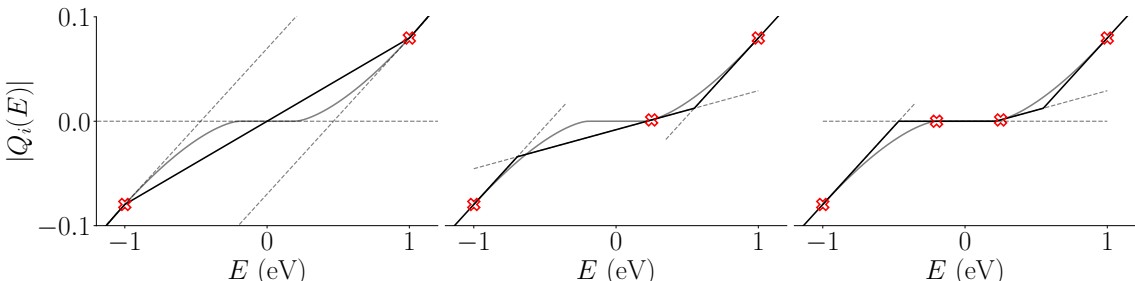

Figure 1: Construction of the piecewise linear ILDOS $\bar{Q}_i(E)$ (black line) from the continuous one $Q_i(E)$ (thin gray line). The red crosses correspond to the initial list of points $E_i^\alpha$. Left: the two tangents (dotted line) do not intersect inside the segment, one interpolates linearly between the two red points. Middle panel: the tangents do intersect, one uses two linear segments (the two tangents) to interpolate between the two red points. Right panel: when a new red point is inserted, the piecewise linear ILDOS $\bar{Q}_i(E)$ is updated.

converge. In this situation, the following, somewhat slower but very stable, "Piecewise Linear Helmholtz algorithm" solves the problem. This algorithm can also be used in cases where the quantum solver provides the ILDOS $Q_i(E)$ but not (its derivative) the LDOS $\rho_i(E)$.

The Piecewise Linear Helmholtz algorithm uses a piecewise-linear-ILDOS $\bar{Q}_i(E)$ that approximates the actual ILDOS $Q_i(E)$. The NHL problem for $\bar{Q}_i(E)$ is solved exactly using the Piece-Wise Newton-Raphson approach (which is now guaranteed to converge because the linearization is exact) and $\bar{Q}_i(E)$ updated (by adding new points) until $\bar{Q}_i(E) = Q_i(E)$ at the solution of the problem. In contrast to Newton-Raphson, this algorithm cannot "overshoot" during an iteration, hence the functional $F(\{U_i\})$ defined in the previous section is guaranteed to decrease. Here, the points $E_i^\alpha$ are not only used to separate smooth regions; they also correspond to a discretization of the ILDOS. The algorithm to construct $\bar{Q}_i(E)$ works as follows (see Fig.1 for an illustration):

- On each point $E_i^\alpha$, we set $\bar{Q}_i(E_i^\alpha) = Q_i(E_i^\alpha)$.

- On each point $E_i^\alpha$, we calculate the tangent $y = Q_i(E_i^\alpha) - \rho_i(E_i^\alpha)(E - E_i^\alpha)$.

- If the tangent at point $E_i^\alpha$ and $E_i^{\alpha+1}$ *do not* interesect inside the segment $[E_i^\alpha, E_i^{\alpha+1}]$ (left panel of Fig.1), we set $\bar{Q}_i(E)$ to simply interpolate linearly between $E_i^\alpha$ and $E_i^{\alpha+1}$:

$$\bar{Q}_i(E) = Q_i(E_i^\alpha) + (E - E_i^\alpha)\frac{Q_i(E_i^{\alpha+1}) - Q_i(E_i^\alpha)}{E_i^{\alpha+1} - E_i^\alpha} \tag{28}$$

- If the tangent at point $E_i^\alpha$ and $E_i^{\alpha+1}$ *do* intersect (middle panel of Fig.1), we set $\bar{Q}_i(E)$ to be equal to the tangents up to the intersection point. To do so we insert an extra temporary point $E_i^{\alpha'}$ at the intersection, in between $E_i^\alpha$ and $E_i^{\alpha+1}$. This may lead to the appearance of redondant points (where there is no slope change). Those points are ignored in the algorithm.

The idea of the algorithm is to solve the piecewise-linear-NLH equation defined by $\bar{Q}_i(E)$ and at the same time refine our description of $\bar{Q}_i(E)$ so it becomes a fair approximation of $Q_i(E)$. We refine $\bar{Q}_i(E)$ by inserting new points $E_\alpha$. The solver works as follows:

1. We initialize the potential on all sites $U_i$. We also initialize the points $E_i^\alpha$. We construct the corresponding piecewise linear ILDOS $\bar{Q}_i(E)$. On each site, we identify the energy interval $\alpha(i)$ such that $U_i \in [E_i^{\alpha(i)}, E_i^{\alpha(i)+1}]$.

2. We solve the linear Helmholtz equation associated with $\bar{Q}_i(E)$. We obtain $U_i'$.

3. For all sites, if $U_i' \in [E_i^{\alpha(i)}, E_i^{\alpha(i)+1}]$ then we set $U_i \rightarrow U_i'$. If $U_i' \leq E_i^{\alpha}(i)$ then the corresponding point has switched to the previous branch. We set $\alpha(i) \rightarrow \alpha(i) - 1$ and $U_i$ to the middle of the previous branch. Likewise, if $U_i' > E_i^{\alpha(i)+1}$ then we switch to the next branch. We set $\alpha(i) \rightarrow \alpha(i) + 1$ and $U_i$ to the middle of the next interval.

4. If we have not switched branch, then the new point $U_i'$ is used to update our piecewise linear ILDOS $\bar{Q}_i(E)$. The value $U_i'$ is added to the list of energies $\{E_i^{\alpha}\}$ cutting the previous interval $[E_i^{\alpha(i)}, E_i^{\alpha(i)+1}]$ into two subintervals $[E_i^{\alpha(i)}, U_i']$ and $[U_i', E_i^{\alpha(i)+1}]$. We reconstruct $\bar{Q}_i(E)$ using this new point (see the right panel of Fig.1 for an example).

5. We repeat steps (2)-(4) until convergence.

In this algorithm, the intervals $E_i^{\alpha}$ are not static, they evolve dynamically along the iterations. Furthermore, since we split the intervals $\alpha(i)$ at the position of the previous iteration energy solution, $\bar{Q}_i(E)$ will be more refined near the actual solution of the non-linear Helmholtz problem. Note that if $Q_i(E)$ is actually piecewise linear, then step (4) above is omitted.

# 5  Numerical examples of solutions of NLH problems

We now turn to practical illustrations of the different algorithms described above on a practical use case. More applications can be found in the other two articles of this series [1, 2], in an application to graphene pn-junction [34] and in simulations of scanning gate microscopy [37]. All the numerics shown here were performed using the open source software PESCADO described in the third article of this series [2].

## 5.1  The ILDOS is actually piecewise linear

We consider an infinitely long hexagonal nanowire. We suppose that it is invariant by translation and therefore the electrostatic modeling is done in two dimensions (for the quantum part it will be important to remember that the problem is 3D, hence has energy bands instead of discrete levels). A back gate (Dirichlet boundary condition at $U_i = -6V$) positioned below the nanowire depletes it while a top gate (Dirichlet boundary condition at $U_i = +2V$) placed over two of the nanowire edges attract electrons into the system. The LDOS vanishes outside the nanowire. A side view of the sample is shown in Fig.2.

We start with the simplest situation where the ILDOS is piecewise-linear with only two branches: an horizontal and a vertical branch. Fig. 2a shows the ILDOS and Fig. 2b the geometry together with the final result (color plot shows the charge on each cell). In fact, this NLH problem corresponds to the PESCA approximation studied in the first article of this series [1].

For this first example we have used the piecewise Linear Helmholtz algorithm. However, since the ILDOS is *actually* piecewise linear, step (4) can be omitted. There is no need to refine something which is already exact. Fig. 2c shows the different iterations until convergence (iteration 7) for an initially random $\alpha(i)$ configuration (iteration 0). We observe that convergence is reached in very few (seven) iterations. Contrary to the Newton-Raphson algorithm, the convergence here is not a continuous process. It is only when all the sites are in the correct branch that the piecewise Linear Helmholtz algorithm has converged (there is no precison criteria). The electrons only accumulate on a single layer of cells, this is a limiation of the PESCA approximation. Indeed, since the density of states is infinite (vertical branch of

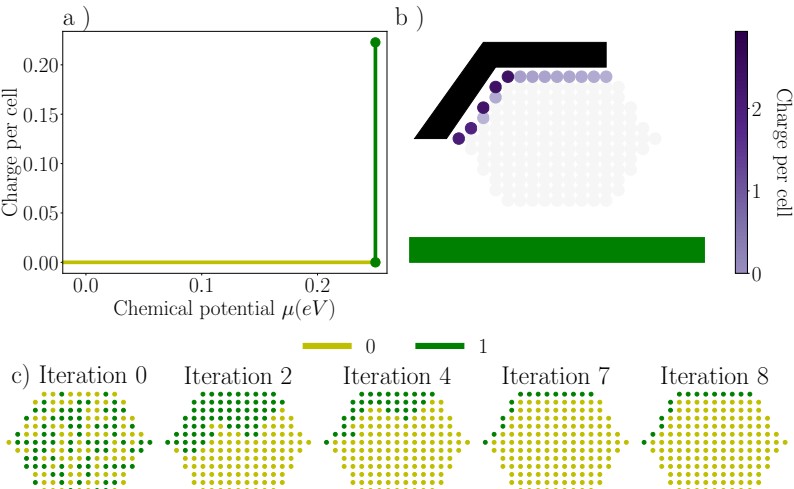

Figure 2: Solving the NLH equation using the piecewise linear Helmholtz algorithm: PESCA. a) Input ILDOS: a piecewise linear ILDOS with two branches. This corresponds to the PESCA approximation. b) Colormap and schematics of a side view of the the hexagonal nanowire with top (black) and back gate (green). The color code corresponds to the charges in each cell of the nanowire. c) Convergence of $\alpha(i)$ on each site versus different iterations. $\alpha = 0$ (yellow, first branch) or $\alpha = 1$ (green, second branch). The initial configuration $\alpha(i)$ is random here.

the ILDOS), the corresponding Thomas-Fermi length vanishes and charge accumulates purely on the surface (purely metallic limit).

In our second example we replace the vertical line of the ILDOS (PESCA) by a line with a finite slope:

$$Q_i(E) = \begin{cases} 0 & \forall & E < 0.25 eV \\ \rho E & \forall & E \geq 0.25 eV \end{cases} \tag{29}$$

This new branch corresponds to the Thomas-Fermi approximation. Fig. 3 shows the results. Convergence is even faster than in PESCA and it is also more acurate as it accounts for a finite density of states in the wire (at no additional computing cost). Thomas Fermi usually leads to a good, quantitative, description of the electronic density. The main difference with PESCA is that the finite density of states means the electric field can now penetrate inside the wire over a finite (Thomas-Fermi) length (see Fig. 3b).

In the last example of this series, we use an ILDOS with three branches. The first describes the valence band, the second the gap and the third the conduction band of a semi-conductor, see Fig.4a. We have,

$$Q_i(E) = \begin{cases} \rho E + \rho \Delta & \forall & E \leq -\Delta \\ 0 & \forall & -\Delta \leq E \geq \Delta \\ \rho E - \rho \Delta & \forall & E \geq \Delta \end{cases} \tag{30}$$

Due to the valence band and the large negative voltage applied to the back gates, it is now possible to attract holes at the lower part of the nanowire (in red, see Fig.4b).

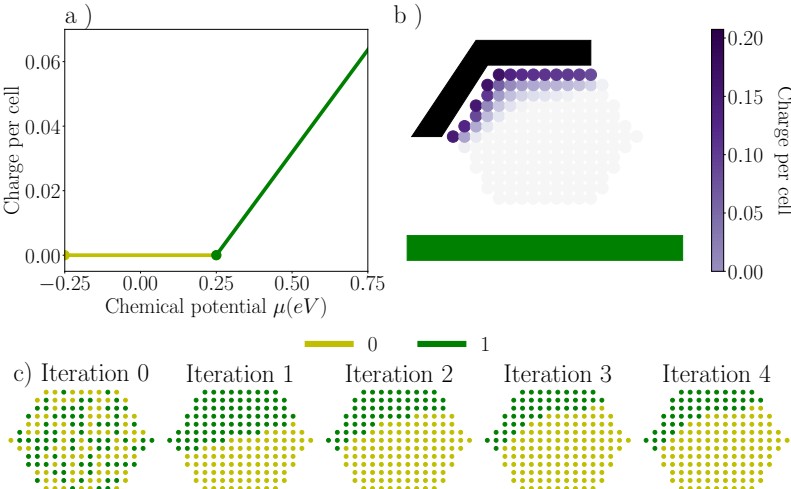

Figure 3: Solving the NLH equation using the piecewise linear Helmholtz algorithm: Thomas-Fermi. a) Input ILDOS: a piecewise linear ILDOS with two branches. This corresponds to the Thomas-Fermi approximation. b) Colormap and schematics of a side view of the the hexagonal nanowire with top (black) and back gate (green). The color code corresponds to the charges in each cell of the nanowire. c) Convergence of $\alpha(i)$ on each site versus different iterations. $\alpha = 0$ (yellow, first branch) or $\alpha = 1$ (green, second branch). The initial configuration $\alpha(i)$ is random here.

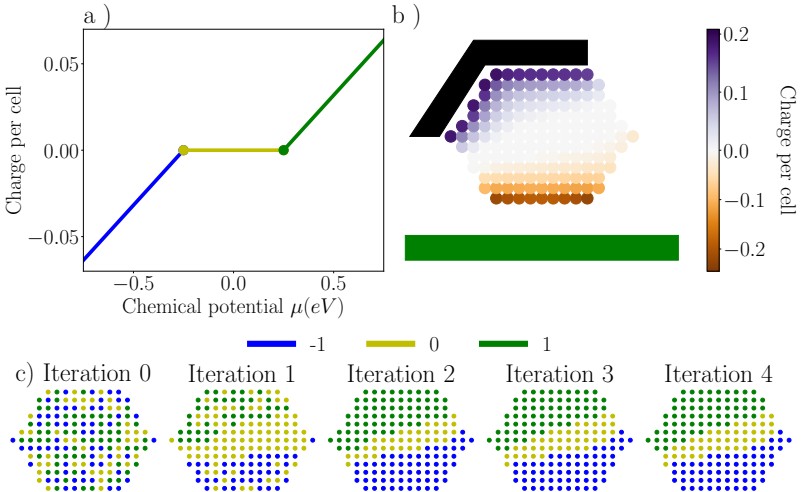

Figure 4: Solving the NLH equation using the piecewise linear Helmholtz algorithm: Valence-Conduction band model. a) Input ILDOS: a piecewise linear ILDOS with three branches. They correspond respectively to the valence band, the gap and the conduction band. b) Colormap and schematics of a side view of the the hexagonal nanowire with top (black) and back gate (green). The color code corresponds to the charges in each cell of the nanowire. c) Convergence of $\alpha(i)$ on each site versus different iterations. $\alpha = -1$ (blue, valence band branch) $\alpha = 0$ (yellow, gap branch) or $\alpha = 1$ (green, conduction band branch). The initial configuration $\alpha(i)$ is random here.

## 5.2   Continuous ILDOS

We now turn to a genuine NLH equation where the ILDOS varies continuously in a non-linear manner. We describe the valence and conduction bands using a free 3D density of states:

$$Q_i(E) = \begin{cases} -a|E + \Delta|^{3/2} & \forall & E \leq -\Delta \\ 0 & \forall & -\Delta \leq E \geq \Delta \\ a|E - \Delta|^{3/2} & \forall & E \geq \Delta \end{cases} \tag{31}$$

This is a Piecewise continuous ILDOS, hence we can use both Piecewise Newton-Raphson and Piecewise Linear Helmholtz algorithms. Fig.5 shows the maximum error for the charge (right) and chemical potential (left) as a function of the number of calls to the linear Helmholtz solver. Both algorithms (green: Newton-Raphson, blue: Piecewise linear solver) converge quickly to the required precision - here set to $\mu = 10^{-10} eV$. The Newton-Raphson is slightly faster ($\sim 10$ iterations compared to $\sim 13$ for the Piecewise linear solver). A typical time to solution on a desktop is 10 seconds for this small problem (5310 sites in total including the 161 "quantum" sites which are treated self-consistently).

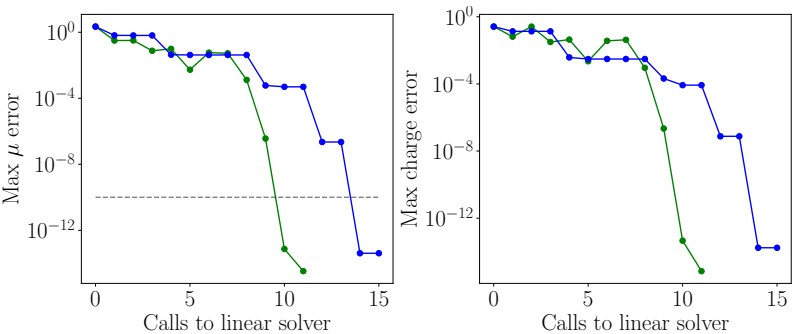

Figure 5:  Error on the chemical potential and charge (max over all sites) as a function of number of calls to the linear Helmholtz equation solver for the continuous ILDOS of Eq.(31). Green: Piecewise Newton-Raphson; Blue: Piecewise Linear algorithm. The dashed grey line of the left figure is the convergence precision criteria for the chemical potential.

Fig.6 illustrates the Piecewise Linear Helmholtz algorithm for this ILDOS. The upper panels show the evolution of the charge in the nanowire at different iterations of the piecewise linear ildos. The middle and lower panels show the corresponding piecewise linear ILDOS $\bar{Q}_i(E)$ on two different representative sites (green and black cross in the upper panel). After just two iterations, the result is indistinguishable from the fully converged result. On each plot in Fig.6 (b) and (c), the small circle correspond to the new solution $U_i'$ obtained after the call to the LH solver. This new solution is used to improve $\bar{Q}_i(E)$. Observe how these new points accumulate close to the final solution (right panel) such that, in fine, $\bar{Q}_i(E)$ is a extremely good approximation of $Q_i(E)$ for $E$ close to the solution $E = U_i$. On the last column the results have been zoomed and show no noticeable difference between $\bar{Q}_i(E)$ and $Q_i(E)$ close to the solution.

## 6   Application to the SCQE problem

We end this paper by solving the full SCQE problem for the nanowire system. The (2D) electrostatic problem is unchanged. For the (3D) quantum problem, we work in the effective mass

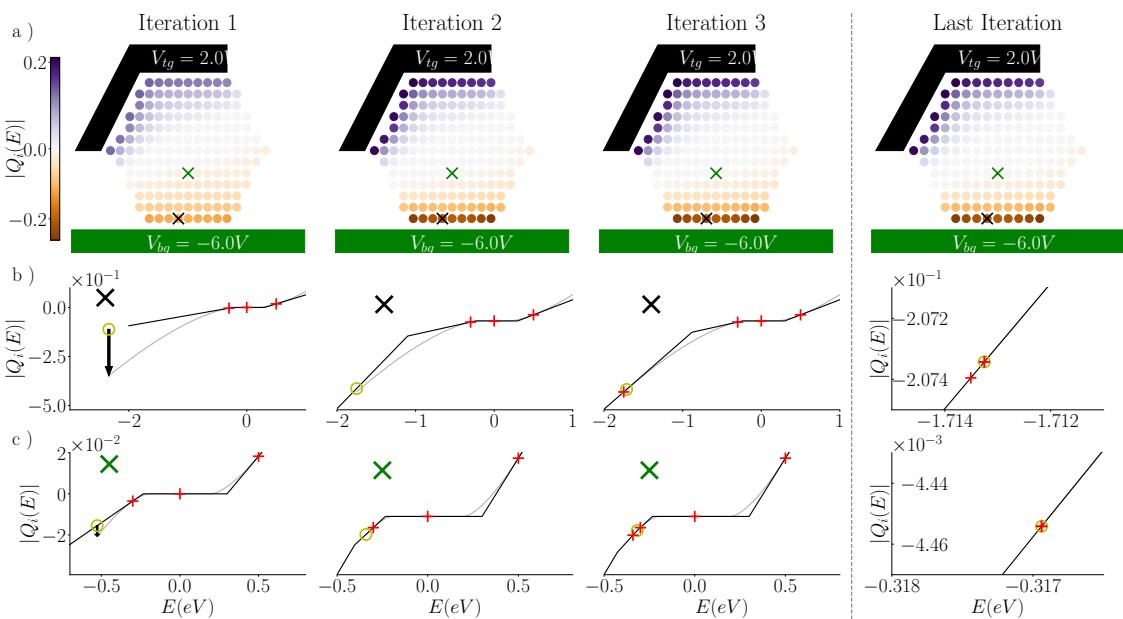

Figure 6: Solving the NLH equation using the piecewise linear Helmholtz algorithm: continuous ILDOS Eq. (31). a) Charge profile for the first three and last iterations. b) Evolution of $\bar{Q}_i(E)$ (black line) for same iterations and for the site tagged by a black cross in a). Thin grey line: continuous ILDOS $Q_i(E)$ that is being approximated. Red crosses: points $E_i^\alpha$ added to the list, yellow circle: value $U_i'$ given by the LH solver. c) Same as b) for the site tagged by a green cross in a)

approximation,

$$H_{3D} = \frac{P^2}{2m^*} - eU(\vec{r}) \tag{32}$$

The system being invariant by translation along the wire direction, the LDOS and ILDOS can be computed semi-analytically as

$$\rho_i(\mu) \quad = \rho^* \sum_\alpha |\Psi_i^\alpha|^2 (\mu - E_\alpha)^{-1/2} \theta(\mu - E_\alpha) \tag{33}$$

$$Q_i(\mu) \quad = \rho^* \sum_n |\Psi_i^\alpha|^2 (\mu - E_\alpha)^{1/2} \theta(\mu - E_\alpha) \tag{34}$$

where $\Psi_i^\alpha$ (resp. $E_\alpha$) is the eigenstate $\alpha$ on cell $i$ (resp. eigenvalue $\alpha$) of the 2D Hamiltonian $H_{2D}$ discretized on the same grid as the one used for the electrostatic (lattice spacing: $a = 5nm$, effective mass: $m^* = 0.067$); the constant $\rho^* = \sqrt{2m^*}/(2\pi\hbar)$ accounts for the 1D density of states along the wire axis. An update of the ILDOS simply amounts to solving $H_{2D}\Psi^\alpha = E_\alpha\Psi^\alpha$ for

$$H_{2D} = -\frac{\hbar^2}{2m^*a^2} \sum_{<ij>} |i\rangle\langle j| - e\sum_i U_i|i\rangle\langle i| \tag{35}$$

where $< ij >$ stands for summation over nearest neighbours. In practice, we perform this calculation using the Kwant package [4].

Figure 7 shows an example of the LDOS and ILDOS of Eq.(33). At every mode opening the ILDOS has a kink - which can destabilize a pure Newton-Raphson algorithm. We will use two approaches to solve the SCQE problem : (i) Piecewise Linear Helmholtz algorithm and (ii) Piecewise Newton Raphson were we split the ILDOS into two, before and after the first mode opens at $\mu = 0$.

Figure 8 shows the charge profile at different voltage regimes for the self-consistent calculations using the Piecewise Linear Helmholtz solver. The first row shows the Thomas Fermi

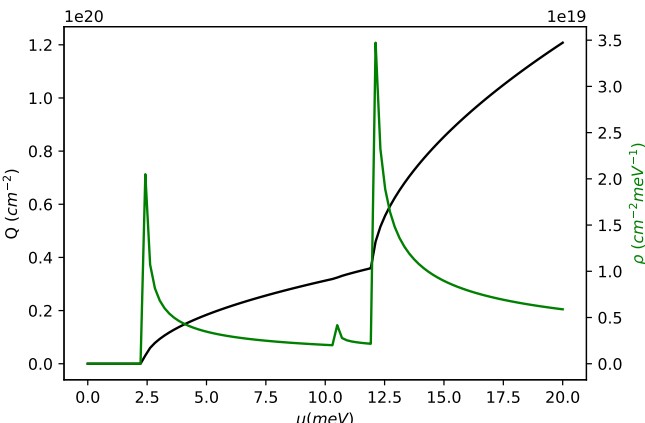

Figure 7: Local density of states (green) and Integrated local density of states (black) for Eq.33. Calculated using Kwant with a system size of 60x60 sites, lattice spacing of 5$nm$ and effective mass of $m^* = 0.067$.

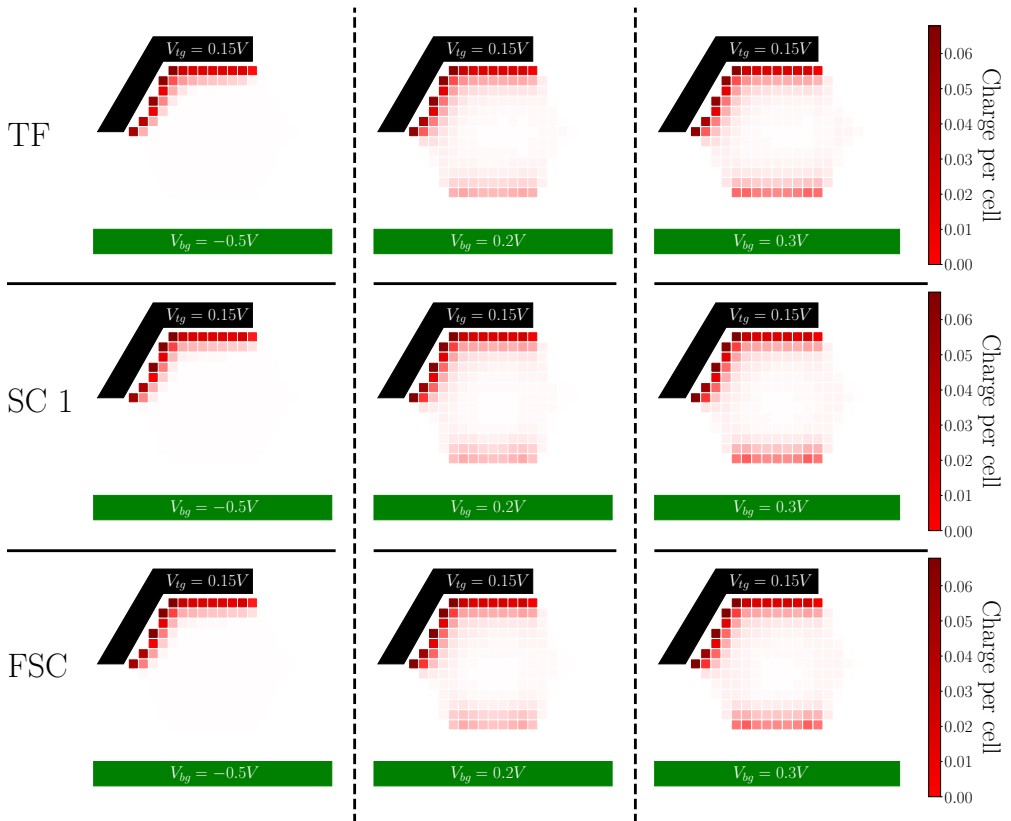

Figure 8: Charge profile for the self-consistent problem defined in Eq.33 under the Thomas-Fermi (TF) approximation, after one iteration of the self-consistent problem (SC 1) and the "full self consistent" converged result (FSC) - top to bottom. The top gate voltage is fixed at $V_{tg} = 0.15V$ and the bottom gate voltages are $V_{bg} = -0.5V$ (left), $V_{bg} = 0.2V$ (middle) and $V_{bg} = 0.3V$ (right). Both NLH solvers converge to machine precision for these parameters.

charge profile, the second row shows the charge after one self-consistent step and the thrid row shows the fully converged self-consistent calculation. Each column corresponds to a dif-

ferent value for the back gate voltage, the top gate voltage is kept constant for all columns. At the level of what can be appreciated with the eye, the results are essentially converged after one update of the LDOS.

For the values of the voltages used in Figure 8 both Piecewise Linear Helmholtz and Piecewise Newton Raphson algorithms converge quickly and to the same result. However, we have identified regimes where the precision of the Piecewise Newton Raphson saturates while the Piecewise Linear Helmholtz converges in a much more robust way (as expected). We ilustrate this point by varying the back gate voltage $V_{bg}$ from $-0.02V$ to $-0.09V$ with a step of $-0.005V$ while we keep the top gate voltage fixed at $V_{tg} = 0.15V$. We find that while the Piecewise Linear Helmholtz converges for every value of $V_{bg}$, the Piecewise Newton Raphson method fails to converge for $V_{bg} \in ]-0.065, -0.04[$. Figure 9 shows the charge and chemical potential error (obtained by taking the max of the error with respect to our most precise result) as a function of the number of calls to the linear Helmholtz solver for $V_{bg} = \{-0.02, -0.05, -0.06\}$ - in green, red and black respectively. For $V_{bg} = \{-0.05, -0.06\}$ Piecewise Newton Raphson oscillates and stagnates at a precision around $10^{-3}$ while the other technique reaches machine precision.

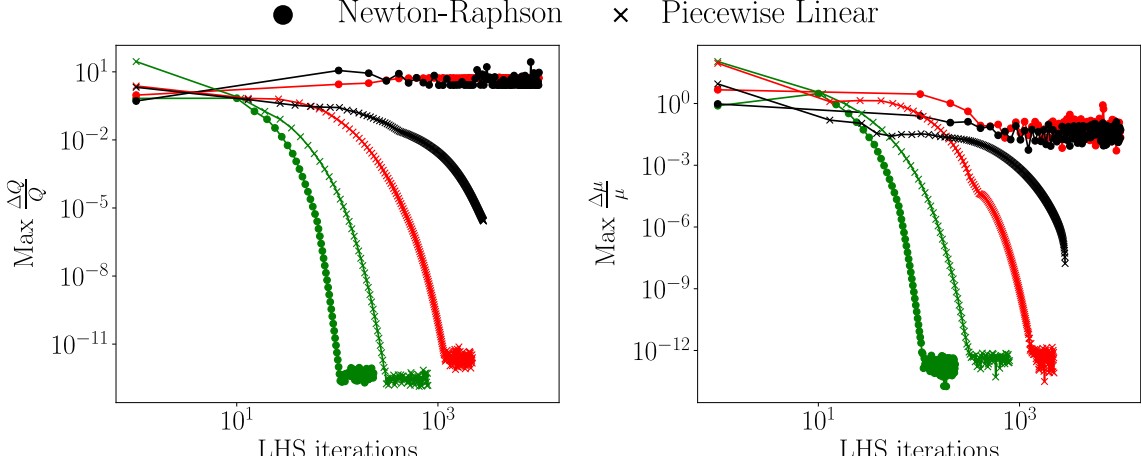

Figure 9: Convergence of the SCQE loop for $V_{bg}$ of $-0.02V$ (green), $-0.05V$ (red) and $-0.06V$ (black). Different symbols correspond to Piecewise Linear Helmholtz algorithm (crosses) and Piecewise Newton Raphson algorithm (circles). We obtain machine precision for Piecewise Linear Helmholtz while the error of the Piecewise Newton Raphson calculation stagnates for $-0.05V$ (red) and $-0.06V$ (black).

## 7 Conclusion

In this paper we have shown a robust route to solve the self-consistent quantum-electrostatics problem. Our approach is based on the combination of

- a finite volume electrostatic solver which guarantees that a *physically correct* problem is solved even when the discretization is coarse.

- a non-linear Helmholtz solver which captures almost all the difficulties of the problem yet can be solved with *provable convergence* (its solution is the unique minimum of a convex functional).

- an explicit treatment of the band edges which are at the origin of most of the non-linearities.

- a simple update scheme of the NLH problem which converges in practice in very few iterations. The self-consistency is not performed on the just the density (standard approach) or even the density and an approximate density of state at the Fermi level (more advanced approaches) but on the *entire* ILDOS versus space and energy.

The main advantage of the above approach is robustness: we have found that we could get the SCQE solution reliably without any complicated tuning of meta parameters of the solver or having to resort to using e.g. a finite temperature in the simulation. A secondary advantage is the precision of the calculation which is, in our experience, not limited by the solver. While the typical voltages applied on the gate lies in the $1eV$ range, a typical Fermi level in a semiconductor is around $1meV$ and one may want to reach sub $100\mu eV$ accuracy. Hence being able to solve the SCQE problem with a large accuracy will be critical for using these tools in quantum nanoelectronic applications.

**Funding information**    X.W. acknowledges funding the European Union H2020 research and innovation program under grant agreement No. 862683, "UltraFastNano" as well as from the Agence Nationale de la Recherche under the France 2030 programme, reference ANR-22-PETQ-0012, the PEPR EPIQ and equbitfly, the ANR TKONDO. X.W. and A.L.S acknowledge funding from the ANR DADDI.

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
