# Peer review of "Electrostatics in semiconducting devices II : Solving the Helmholtz equation"

_SciPost Physics_

## Round 1 · Referee Report · Anonymous (Referee 1) · 2025-8-24

Report

The authors present a robust numerical approach to the self-consistent quantum–electrostatics (SCQE) problem, combining a finite-volume electrostatics solver with a nonlinear Helmholtz (NLH) solver and an explicit treatment of band edges. The method is clearly motivated, addresses well-known shortcomings of previous linearization and predictor–corrector approaches, and appears to converge reliably without delicate parameter tuning.

The manuscript is well written and provides a clear exposition of prior work and of the present contribution. While I find the work sound and of potential interest to the community, I encourage the authors to clarify a few points. In particular, it would be useful to more explicitly contrast their approach with prior nonlinear Poisson or Newton–Raphson schemes, to provide quantitative benchmarks, and to discuss practical limits of the method. Some additional algorithmic details (e.g. pseudocode or a flowchart of the self-consistency loop) would make the implementation more reproducible. Finally, short outlook on possible extensions (e.g. more complex geometries or disorder) would strengthen the paper.

Minor comment: on page 9, the notation for the energy should read
$E^{\alpha(i)}_i$ not $E^{\alpha}_i(i)$. Please correct this typo.

Overall, this is a strong and well-presented manuscript. I recommend publication after minor clarifications and improvements in presentation.

Recommendation

Ask for minor revision

---

## Round 1 · Referee Report · Anonymous (Referee 2) · 2025-9-5

Report

In this second part of their series of articles about electrostatics in nanostructures, the Authors describe a robust recipe to solve the Schrödinger-Poisson problem, and showcase its application to the case of a nanowire with hexagonal cross-section.

On the technical level, the research presented here represents a remarkable advance. Furthremore, the article is very clearly written. Given that this type of calculations are becoming more and more important and widespread, I can expect that the method proposed will be of high impact.

I have some minor objections related to clarity:

1) The relation with the PESCA method presented in the first installment of the series is not made very clear. In fact, I think that a reader interested in using the method to tackle a Schrödinger-Poisson problem may skip the first article and study directly this second one. On one hand, this is good since it makes this article self-contained. On the other, it would be better to explicitate the relation between the first and second article of the series. (Note: this comment of mine mirrors what I wrote in my report for the first article).

2) In Section 5.2 the Authors say their simulations has "5320 sites in total including the 161 quantum sites which are treated self-consistently". I assume that the 161 sites are those shown explicitly in e.g. Figure 6, forming the hexagonal semiconducting wire cross-section. Where are all the other ~5000 sites and why so many are needed? I hope the Authors can provide more details on this aspect of the simulation, as an opportunity to also explain some criteria on how to choose the discretization grid in practice. It is often this nitty-gritty details that make a method successful or not and so, even if general receipre cannot be provided, examples help.

3) The Authors mention several times that the electrostatic problem is non-local. However, in the discretized version described in Sec. 2, the only non-zero matric elements of the capacitance matrix are diagonal ones and nearest-neighbor ones, since it is essentially a discretized Laplacian. This comes from the first-order approcimation of the gradiant. I suppose it is this feature that makes the problem convergent, as opposed to the full SCQE problem? If this is so, I would emphasize this point.

Recommendation

Ask for minor revision

---

## Round 1 · Referee Report · Anonymous (Referee 3) · 2025-9-8

Report

This paper describes a clever way to numerically handle the self-consistent quantum electrostatics (SCQE) problem by mapping the original problem onto a non-linear Helmholtz(NLH) equation. The key is to replace the integrated local density of states at each position by a much simpler piecewise linear function that is locally refined at each iteration. The solution of this approximate problem converges to that of the SCQE as the local density of states is locally refined. The authors show that gradient descent approach must arrive at a unique solution.

This approach seems quite useful for solving non-local problems with non-linearities. I have a couple of questions: 1. Why can NLH not overshoot during an iteration? (Page 10) 2. Do the authors have insight on when the Newton-Raphson scheme is expected to fail or succeed? (Page 9 and Figure 9)

Issues: 1. NLH and NHL are used interchangeably throughout the paper 2. Trivial notes (spell check may be helpful) 1. Redondant -> Redundant 2. Eponym -> eponymous 3. semi-definite positive -> positive semi-definite

Recommendation

Publish (meets expectations and criteria for this Journal)

---

## Editorial Decision

awaiting_resubmission